# Kolmogorov–Arnold Network Model Integrated with Hypoxia Risk for Predicting PD-L1 Inhibitor Responses in Hepatocellular Carcinoma

**DOI:** 10.3390/bioengineering12030322

**Published:** 2025-03-20

**Authors:** Mohan Huang, Xinyue Chen, Yi Jiang, Lawrence Wing Chi Chan

**Affiliations:** 1Department of Health Technology and Informatics, The Hong Kong Polytechnic University, Hong Kong SAR, China; mo-han.huang@connect.polyu.hk (M.H.); xinyue24.chen@connect.polyu.hk (X.C.); 2The Second Affiliated Hospital, School of Medicine, The Chinese University of Hong Kong, Shenzhen 518000, China; jacky_1001@163.com

**Keywords:** hepatocellular carcinoma, hypoxia, immunotherapy response, Kolmogorov–Arnold network, support vector machine

## Abstract

Hepatocellular carcinoma (HCC) is a leading cause of cancer-related deaths, with immunotherapy being a first-line treatment at the advanced stage and beyond. Hypoxia plays a critical role in tumor progression and resistance to therapy. This study develops and validates an artificial intelligence (AI) model based on publicly available genomic datasets to predict hypoxia-related immunotherapy responses. Based on the HCC-Hypoxia Overlap (HHO) and immunotherapy response to hypoxia (IRH) genes selected by differential expression and enrichment analyses, a hypoxia model was built and validated on the TCGA-LIHC and GSE233802 datasets, respectively. The training and test sets were assembled from the EGAD00001008128 dataset of 290 HCC patients, and the response and non-response classes were balanced using the Synthetic Minority Over-sampling Technique. With the genes selected via the minimum Redundancy Maximum Relevance and stepwise forward methods, a Kolmogorov–Arnold Network (KAN) model was trained. Support Vector Machine (SVM) combined the Hypoxia and KAN models to predict immunotherapy response. The hypoxia model was constructed using 10 genes (IRH and HHO). The KAN model with 11 genes achieved a test accuracy of 0.7. The SVM integrating the hypoxia and KAN models achieved a test accuracy of 0.725. The established AI model can predict immunotherapy response based on hypoxia risk and genomic factors potentially intervenable in HCC patients.

## 1. Introduction

Hepatocellular carcinoma (HCC) represents the predominant subtype of primary liver cancer and one of the leading causes of cancer-related mortality globally. The majority of HCCs are diagnosed at advanced stages [1]. During the disease trajectory, over 50–60% of HCC patients undergo systemic therapy, which has emerged as the mainstay treatment modality for patients presenting with unresectable tumors [2].

Immunotherapy has emerged as a groundbreaking addition to the armamentarium of systemic therapies. The immune checkpoint inhibitor (ICI), Atezolizumab (PD-L1 inhibitor), in combination with Bevacizumab (VEGF antibody), has received FDA approval as a first-line treatment for unresectable HCC [3]. Atezolizumab binds to PD-L1, disrupting the PD-1/PD-L1 axis and enhancing the cytotoxic activity of anti-tumor T cells. However, immunotherapy faces significant challenges, such as lower response rates, drug resistance, and side effects [4]. Specifically, the objective response rate remains around 30%, and the 3-year overall survival rate falls short of 50% [5,6]. A significant proportion of patients fail to derive benefit from anti-PD-1/PD-L1 therapy; this treatment is associated with high costs and the potential for adverse events. Comprehending the long-term treatment outcomes of immunotherapy in HCC may offer insights to tailor treatments for individual patients [7].

The tumor immune microenvironment (TIME) plays a pivotal role in mediating the response to ICIs and has a profound influence on both the efficacy of immunotherapeutic interventions and tumor progression [8]. The tumor microenvironment (TME) of HCC is largely immunosuppressive [9]. Hypoxic TIME is a prevalent feature of HCC solid tumors, which is caused by the rapid proliferation of HCC cells combined with an inadequate vascular oxygen supply [10]. Such an environment triggers the activation of various oncogenic markers and promotes the upregulation of PD-L1 expression. This enhances the tumor’s resistance to cytotoxic T-lymphocytes (CTLs) and contributes to immunotherapy tolerance and resistance [11].

Current genomic research primarily utilizes two approaches: machine learning (ML)and bioinformatics. ML employs algorithms to find complex interactions among genomic factors, achieving high predictive accuracy when supported by genomic sequencing data connected to clinical information [12]. Bioinformatics relies on statistical methods such as differential gene expression analysis, enrichment analysis, and regression analysis to identify potential genes, pathways, and molecular mechanisms [13]. Both approaches have limitations in capturing the intricate interactions between high-dimensional genomic data and TME, particularly the critical role of hypoxia in regulating immune responses. Therefore, traditional models often fail to integrate diverse biological features that contribute to therapeutic resistance, thereby limiting their clinical applicability.

Kolmogorov–Arnold Networks (KANs) represent a recently introduced, novel deep-learning model framework. KANs are constructed based on the Kolmogorov–Arnold theorem, using parametrized splines as the learnable activation functions on edges, instead of inputting the weighted sum to the fixed activation functions as Multi-Layer Perceptrons (MLPs). MLPs are feedforward neural networks with multiple layers of neuron connections, which can handle certain nonlinear relationships. But MLPs are always falling into local optima, have limited capabilities in processing high-dimensional data, and have poor interpretability. The modeling performance of KANs is comparable to and, on small-scale tasks, even better than that of MLPs. Compared with traditional deep learning networks, the intuitive representation of KANs facilitates high interpretability, and they can clearly present the mapping relationship between input features and output results [14]. The application of KANs has a better performance in handling and analyzing high-dimensional biomedical data [15]. This study used a KAN to model the effect of genomic factors on the immunotherapy response in HCC, which facilitates an in-depth exploration of genomic features associated with immune response. Hypoxia plays a critical role in the TME, influencing both tumor immune evasion and treatment resistance. Existing predictive models for HCC immunotherapy response often overlook the interaction between hypoxia and immune regulation. By integrating hypoxia-related factors with KAN, the complex interplay between the TME and high dimensional genomic data has been captured.

In this study, we trained the Cox regression model on multiple publicly available clinical and cell experiment datasets to predict the hypoxia risk. The genes used in this model were selected using differential expressions. Meanwhile, we trained the KAN model on a different clinical dataset to predict immunotherapy response. We used a support vector machine (SVM) to integrate the outputs of two heterogeneous models into a single score. This approach effectively combines their advantages by considering both the hypoxia risk in the TME and using highly interpretable immune response features.

## 2. Methods

The study design is illustrated in Figure 1.

### 2.1. Data Collection and Preprocessing

Bioinformatics analysis, modeling, and validation were performed on publicly available datasets: EGAD00001008128, from the European Genome-Phenome Archive (EGA) (https://ega-archive.org/datasets/ downloaded on 10 October 2023), GSE41666, and GSE14520, from the National Center for Biotechnology Information (NCBI) database (https://www.ncbi.nlm.nih.gov/—accessed on 26 January 2022 and 5 January 2022, respectively)—as well as The Cancer Genome Atlas Liver Hepatocellular Carcinoma (TCGA-LIHC) dataset from the TCGA database (https://www.cancerimagingarchive.net/collection/tcga-lihc/, on 10 January 2024).

#### 2.1.1. Clinical Datasets

The EGAD00001008128 dataset includes RNA sequencing data of 290 HCC patients who underwent treatment with Atezolizumab or Atezolizumab + Bevacizumab following biopsies. The responses to the treatment were categorized into positive (90 patients) and negative (200 patients). The dataset was divided into training and test sets for immunotherapy response prediction model development. As the non-response class constitutes the majority of cases, a 9:1 split was first applied to the non-response cases. Of the 200 non-response cases, 20 were randomly selected for the test set, alongside 20 of the 90 response cases. The training set thus consists of 180 non-response cases and 70 response cases. The GSE14520 dataset consists of microarray data of HCC tumor tissues compared to adjacent normal tissues of 214 patients. The Cancer Genome Atlas (TCGA) Liver Cancer (LIHC) dataset consists of gene expression profiles (FPKM values) and survival data of 367 patients.

#### 2.1.2. Cell Culture Datasets

The GSE41666 dataset includes gene expression profiles of the HepG2 cell line cultures under the hypoxic (0% O_2_) and normoxic (21% O_2_) conditions for 24 h, generated using a microarray platform. Each condition in GSE41666 was replicated three times to ensure biological reproducibility.

#### 2.1.3. Pre-Processing of RNA Sequencing Raw Data

For RNA sequencing data, the dataset EGAD00001008128 was processed using a standard processing pipeline that includes (i) FastQC for initial quality assessment, (ii) Fastp for the trimming of low-quality bases and removal of adapters, (iii) HISAT2 for the alignment of cleaned reads to the human reference genome (GRCh38.104 from NCBI), and (iv) FeatureCounts for gene expression quantification, all ensuring the data integrity, quality, and reliability for comprehensive analysis. Expression matrices were then formed [16].

To account for low quality sequencing results, gene filtering methods adopted by relevant studies were applied. For the EGAD00001008128 dataset, the median normalized counts-per-million (CPM) and the coefficient of variation (CV) of values were calculated for each gene, with 25th percentiles used as thresholds. Genes were excluded if they fell below these thresholds or were expressed in less than 75% of the samples [2].

#### 2.1.4. Normalization of Gene Expression Data

Gene expression data from the GSE41666 dataset, generated using microarray technology, was processed by averaging the expression levels of multiple probes corresponding to each gene, resulting in an expression matrix with unique gene symbols. The data underwent Variance Stabilizing Normalization (VSN) and was subsequently standardized to a normal distribution, N (0, 1).

For the RNA sequencing datasets EGAD00001008128, our study preprocessed the datasets using the DESeq2 normalization function (deseq2_norm) in Python (3.8.18)/R (4.1.3). This function adjusts for sequencing depth and technical variations among samples by applying sample-specific scaling factors to the expression matrices.

To rescale the expression matrix of EGAD00001008128 into a standard range for further analysis, this study used the StandardScaler function of the sklearn Python package to fit a scaler concerning the mean and standard deviation of the expression profile of each gene in the training set. Both training and test sets were standardized to the normal distribution, N (0, 1), using the fitted scaler. The dataset includes the immunotherapy strategy: Atezolizumab monotherapy (coded by 0) or its combination with Bevacizumab (coded by 1). This variable was then standardized alongside the gene expression data in both training and test sets.

### 2.2. Feature Selection

#### 2.2.1. Differential Expression Analysis

Differential expression analysis (DEA) was performed on the EGAD00001008128, GSE41666, and GSE14520 datasets based on t-test q-values and fold change (FC) criteria. *p*-values were calculated for each gene to determine statistical significance, and q-values were derived using the Storey–Tibshirani method for the GSE41666 and GSE14520 microarray data, and the Benjamini–Hochberg method for the EGAD00001008128 RNA sequencing data to control the false discovery rate (FDR). Genes were considered differentially expressed (DEGs) if they met the criteria of q-value < 0.05 and FC thresholds for upregulation or downregulation.

#### 2.2.2. Enrichment Analysis

Venn diagrams were generated using the Venny 2.1 platform (https://bioinfogp.cnb.csic.es/tools/venny/, accessed on 25 May 2024) to visualize the intersection of DEGs from the GSE41666 dataset with those from the EGAD00001008128 and GSE14520 datasets. Overlapping genes identified across these datasets were subsequently categorized into two distinct groups: IRH (immunotherapy response to hypoxia) genes and HHO (HCC-Hypoxia Overlap) genes. The resulting HHOs have been reported in our previous study [13].

### 2.3. Model Development

#### 2.3.1. Oversampling of Imbalanced Data

Balanced class distributions are known to improve model predictive performance. In the training set of the EGAD00001008128 dataset, a significant class imbalance was observed, with 70 responders and 180 non-responders. To address this, our study performed the Synthetic Minority Over-Sampling Technique (SMOTE), as a previous study indicated that this approach could improve the performance of the classifier for the minority class [17]. SMOTE was applied as an initial preprocessing step. The SMOTE function from the imbalanced-learn library in Python was used to process the small sample size in the imbalanced dataset. SMOTE generates synthetic samples by interpolating between the existing data points in the minority class, i.e., responders in this case, based on their nearest neighbors, while maintaining key characteristics of gene expression profiles and the Bevacizumab treatment option. This process resulted in a balanced training set, with 180 samples in each class. This is achieved by interpolating between an example and its K-nearest neighbors, which could prevent overfitting and enable the classifier to better identify decision boundaries between different classes [18].

#### 2.3.2. Hypoxia Scoring Model Associated with Drug Response

To develop a hypoxia scoring model associated with drug response, gene expression (FPKM values) and survival data from HCC patients in the TCGA Liver Cancer (LIHC) database were analyzed. First, samples with missing survival time or a survival time of 0 days were excluded, resulting in a final set of 367 clinical samples used for training the hypoxia score model. After that, univariate Cox regression analysis was performed on HHOs, followed by application of the Least Absolute Shrinkage and Selection Operator (LASSO) algorithm. Ten-fold cross-validation was performed using the R glmnet package 4.1-8 to determine the optimal parameter λ and the hypoxia-related genes. The results of the analysis were reported in our previous study [19]. Subsequently, hypoxia-related genes were combined with immunotherapy response to hypoxia (IRH) genes to construct the hypoxia scoring model. Ultimately, a proportional hazards regression model was fitted using the phreg function from the Statsmodels library, and the regression coefficients for each gene in the hypoxia scoring model were calculated based on the TCGA dataset. The hazard function *λ*(*t*) is an indicator used to evaluate the performance model. The baseline hazard function *λ*_0_(*t*) represents the risk level when all covariates are zero. The coefficients *β_i_* of the covariates reflect their impact on risk, with positive values indicating increased risk and negative values indicating reduced risk.$$\lambda \left(t\right)=\lambda_{0}(t)\times exp(\beta_{1}\times X_{1}+\beta_{2}\times X_{2}+...\beta_{n}\times X_{n})$$

#### 2.3.3. Model Validation

To address the challenges of current clinical techniques in measuring intratumoral hypoxia, the model was applied to the standardized EGAD00001008128 dataset to evaluate hypoxia risk in HCC patients. The hypoxia risk scores were then incorporated into the feature set for further analysis.

#### 2.3.4. KAN Model Architecture and Training

The Minimum Redundancy Maximum Relevance (mRMR) algorithm was applied to rank genes by their correlation with immunotherapy response, selecting the top 50 most relevant feature genes. Stepwise Forward Selection was then used to iteratively add feature genes to the model, focusing on those with the highest predictive power, which helped simplify the model and reduce overfitting.

The Kolmogorov–Arnold Network (KAN) has comparable mathematical foundations but better accuracy and interpretability than Multi-Layer Perceptron (MLP). Instead of the learnable weights in MLP, the parameters of the activation functions of KAN are adjusted in the training process, leading to a simpler but more precise model for learning limited data. The model prediction, *f*(*x*), is defined as a double summation: The inner summation, $\sum_{i}^{n}\phi_{0,j,i}\left(x_{i}\right)$, transforms each feature *x_i_* using a feature-specific function *ϕ*_0*,j*,*i*_. The outer summation, $\sum_{j}^{m}\phi_{1,j}$, combines these transformed features across all m predictors via *ϕ*_1,*j*_. This structure allows KAN to capture complex feature interactions with improved precision and interpretability.$$f\left(x\right)=f\left(x_{1},\ldots ,x_{n}\right)=\sum_{j}^{m}\phi_{1,j}\left(\sum_{i}^{n}\phi_{0,j,i}\left(x_{i}\right)\right)$$

#### 2.3.5. SVM Model

A Support Vector Machine (SVM) model with a radial basis function (RBF) kernel was employed to predict immunotherapy response, using inputs derived from the KAN output and hypoxia model output. The model was implemented using the Sklearn.svm module, with parameters set to ’gamma = 0.2’ and ’C = 0.1’. The RBF kernel controls the feature mapping through the parameter *σ*, which determines the spread of the kernel and projects features into a higher-dimensional space. In the feature representation, variables *x* and *x_c_* represent genomic features, while the Euclidean distance *||x − x_c_||*, which quantifies the similarity between a sample and the central feature *x_c_*, was incorporated into the RBF kernel. This approach captures the proximity of a sample to the characteristics of responders. By integrating these features, the SVM model effectively identifies interactions between genomic characteristics and hypoxia scores, enhancing the accuracy of immunotherapy response prediction.$$\varphi (x,x_{c})=exp(-\frac{{\left||x-x_{c}|\right|}^{2}}{2\sigma^{2}})$$

## 3. Results

### 3.1. Identification of IRH and HHO Genes from Public Dataset

In the EGAD00001008128 dataset, 84 upregulated DEGs were identified between immunotherapy responders and non-responders using thresholds of q-value < 0.05, FC > 1.999 (upregulation), and FC < 1/1.999 (downregulation) (Figure 2a). To account for variation across groups, overlap was adjusted for greater significance. In the GSE41666 dataset, FC thresholds were set at >1.3202, identifying 300 upregulated hypoxia-related genes (HRGs) for overlap analysis with the EGAD00001008128 dataset (Figure 2b). FC thresholds were adjusted to >1.277 and <1/1.370, identifying 400 upregulated and 400 downregulated genes for overlap analysis with the GSE14520 dataset (q-value < 0.05, FC > 1.139 and <1/1.1276) (Figure 2c). These adjustments and results have been reported in a prior study.

Analysis of the EGAD00001008128 and GSE41666 datasets identified three overlapping genes (PMAIP1 (NOXA), CD3D, and CD2) as immunotherapy response to hypoxia (IRH) genes (*p*-value = 0.026) (Figure 3a). PMAIP1 (NOXA) has been consistently related to hypoxia in previous studies. Further analysis of the GSE41666 and GSE14520 datasets identified 52 statistically significant overlapping genes classified as HCC-Hypoxia Overlap (HHO) genes (*p*-value = 0.0032) (Figure 3b).

### 3.2. Hypoxia Scoring Model

Our study established a hypoxia risk-scoring model based on 10 genes. Among these, 9 genes were identified from HHOs. These genes were first analyzed using univariate Cox regression based on FPKM data from the TCGA-LIHC dataset, and those with significant associations with survival were further refined through the LASSO algorithm and cross-validation (Figure 4). Based on the FPKM data from the TCGA-LIHC dataset, these genes were first analyzed using univariate Cox regression analysis. Subsequently, LASSO regression was carried out for feature selection to optimize the model and avoid overfitting. Figure 4a shows the results of the Cox proportional hazards regression analysis, including *p*-values, hazard ratios (HR), and their 95% confidence intervals (CI). The *p*-values of the 21 candidate genes were relatively low, indicating their statistically significant association with survival. Genes with HR > 1 (e.g., *PHLDA2*, *DLGAP5*, *CENPA*, *HMMR*, *KIF20A*) were associated with poorer prognosis, whereas genes with HR < 1 (e.g., *N4BP2L1*, *UPB1*, *AFM*) were associated with better prognosis. Figure 4b shows the LASSO regression variable selection process, where the optimal λ value was determined using cross-validation. By penalizing regression coefficients, LASSO reduced the number of variables and finally selected nine key genes for the model. The remaining gene, *PMAIP1* (NOXA), was derived from the IRHs. After incorporating *PMAIP1* (NOXA) into the model, the final set of 10 genes was subjected to an additional Cox regression analysis using the TCGA-LIHC FPKM dataset. This step ensured that all genes in the model were significantly associated with hypoxia-related survival outcomes. The model calculates a weighted combination of gene variables, where each gene’s expression is assigned a coefficient representing its impact on survival. Positive coefficients indicate increased risk, while negative coefficients suggest reduced risk (Table 1). The final key genes in the model include *PHLDA2*, *DLGAP5*, *N4BP2L1*, *CENPA*, *UPB1*, *CABYR*, *AFM*, *HMMR*, *KIF20A*, and *PMAIP1*. This formula was applied to the EGAD00001008128 dataset to calculate hypoxia risk scores for each patient, facilitating patient stratification and further analysis.$$\begin{array}{cc} h(t|X)=h_{0}(t)exp & (0.1497\times PHLDA2+0.0757\times DLGAP5-0.2318\times N4BP2L1\\ & +0.0990\times CENPA-0.0584\times UPB1+0.1509\times CABYR\\ & -0.0022\times AFM+0.3139\times HMMR+0.0587\times KIF20A\\ & -0.1203\times PMAIP1) \end{array}$$

### 3.3. Feature Selection and KAN Model

Our study applied the mRMR algorithm to rank genes based on their relevance to immunotherapy response and inter-redundancy. The top 50 genes were initially selected from the EGAD00001008128 dataset. These 50 genes were further refined using the KAN model, which identified 11 genes most strongly associated with immunotherapy response: *TLR8*, *SCHIP1*, *ZNF729*, *ATG9B*, *BAMBI*, *OXSR1*, *ZNF564*, *CCR8*, *ADAM23*, *RRN3P3_1*, and *RDH14*. A training dataset was constructed by combining these 11 selected genes with Bevacizumab usage as features. Both the training and validation datasets were converted into PyTorch 2.6 tensors, ensuring data completeness by verifying and correcting any missing values. The model used a classification threshold of 0.5, binarizing predictions for immunotherapy response. In our study, we conducted six training rounds, each defined as a batch of steps, with varying numbers of grid intervals ranging from three to eight. The number of grid intervals determines the resolution and flexibility of the spline activation functions in the KAN. For model optimization, we employed the L-BFGS algorithm with a learning rate of 1. The maximum number of iterations per optimization process was set to 20, with a maximal function evaluation limit of 25. The termination criteria were defined by a first-order optimality tolerance of 1 × 10^−7^ and a function tolerance of 1 × 10^−9^ in Table 2.

The KAN model achieved an accuracy of 0.9361 on the training set, demonstrating strong performance in binary classification. The trained KAN model was then tested on a validation dataset using the same classification threshold of 0.5. The test set achieved an accuracy of 0.7 (Figure 5). KAN scores from both the training and test datasets were incorporated into the feature columns for further analysis and validation.

### 3.4. SVM Model for Immunotherapy Response Prediction

A support vector machine (SVM) model was trained using the KAN score and hypoxia scores as features to classify immunotherapy response in the EGAD00001008128 dataset. The training set included these two features, along with immunotherapy response results. The model achieved an accuracy of 0.9361 on the training set. When evaluated on the test set, the model achieved an accuracy of 0.725 (Figure 6). The SVM model used a radial basis function (RBF) kernel with parameters gamma = 0.2 and C = 0.1. These results demonstrate that the combination of KAN and hypoxia scores provides strong predictive power in immunotherapy response.

## 4. Discussion

Despite advancements in hepatocellular carcinoma (HCC) treatment, the prognosis for patients with advanced HCC remains poor. Current systemic therapies, including immune checkpoint inhibitors, provide limited benefits for many patients. The high mortality rate of advanced HCC continues to be a major public health challenge. The lack of reliable tools to predict immunotherapy responses further complicates treatment decisions. Therefore, there is an urgent need to develop a predictive model for advanced HCC patients. This model would guide immunotherapy strategies, improve survival outcomes, and optimize clinical decision-making.

In recent years, the relationship between hypoxia-related genes and cancer progression, immune evasion, and treatment resistance has been widely studied. These genes have been used to develop predictive models for survival outcomes and drug resistance in various cancers [20]. The liver, as one of the most metabolically active and hypoxia-sensitive organs, is deeply involved in oxygen homeostasis, tumor metabolism, and immune regulation [21,22]. This study used hypoxia-related and response-related genes to construct a predictive model for immunotherapy response in HCC, providing a novel finding for linking hypoxia with immunotherapy outcomes.

We developed a prediction model for PD-L1 inhibitor response in HCC, an important issue in cancer immunotherapy. This study uses the Kolmogorov–Arnold Network (KAN) integrated with the hypoxia risk scoring model. The hypoxia risk scoring model was constructed using 10 genes. The HCC-Hypoxia Overlap gene set was identified by overlapping the GSE41666 and GSE14520 datasets through Cox regression and LASSO analysis. Nine genes were selected from the HHO gene set that showed strong performance in stratifying patients based on hypoxia-associated survival outcomes. NOXA was selected from the IRH (Immunotherapy Response to Hypoxia) gene set, derived from the intersection of the EGAD00001008128 and GSE41666 datasets. Based on the selected hypoxia-related genes, we analyzed data from the TCGA-LIHC dataset to establish a hypoxia scoring risk model associated with immune response.

NOXA has been well-documented in recent studies for its role in hypoxia and immunotherapy. Hypoxia-induced HIF-1α has been shown to upregulate NOXA mRNA and protein expression by binding to the hypoxia response element (HRE) at −1275 bp within the NOXA promoter, independently of p53 [23,24]. NOXA, a pro-apoptotic BH3-specific protein, plays a key role in tumor cell apoptosis. Studies have shown that NOXA knockout reduces CAR T cell-mediated tumor apoptosis, leading to resistance against CAR T cell therapy [25]. In colorectal cancer, PTEN and NOXA expression are strongly correlated, with high PTEN levels linked to tumor regression [26]. These findings highlight NOXA as a critical regulator of anti-tumor responses across different treatments. Our study further found NOXA expression to be upregulated in hypoxic HepG2 cell lines and in immunotherapy responders, emphasizing its potential as a therapeutic target. Future studies are required to explore whether modulating NOXA expression can enhance responses to immunotherapy.

Subsequently, we developed the KAN model to predict PD-L1 inhibitor response in HCC, demonstrating strong predictive performance. The KAN model contributed significantly to the prediction model by refining the feature set through SMOTE and mRMR. Of the 50 genes initially selected by mRMR, 11 were identified as the most relevant predictors of immunotherapy response. The KAN model achieved high accuracy in both the training set (0.9361) and the test set (0.7), demonstrating its ability to generalize and perform well on unseen data. Traditional immunotherapy predictive models often rely on single biomarkers or clinical features, failing to fully consider the complexity of the TME. For example, tumor mutational burden (TMB)-based predictive models exhibit significant variability in predictive performance across different cancer types and do not account for microenvironmental factors such as hypoxia [27,28]. Compared to traditional models, the KAN model’s adaptive activation functions enabled it to handle the nonlinear and complex interplay between gene expression and immune response more efficiently. This advantage is particularly relevant in HCC, where hypoxia and immune modulation are highly interconnected. Our findings highlight the potential of KAN models for clinical application in personalized medicine.

The integration of hypoxia scores and KAN-derived feature scores in the SVM model showed promising results. Moreover, this combined approach had potential for predicting PD-L1 inhibitor response in HCC. The training set of EGAD00001008128 incorporated these two features alongside immunotherapy response data, achieving an accuracy of 0.9361. When applied to the test set, the model maintained a discriminative performance with an accuracy of 0.725, highlighting its ability to generalize to unseen data. Hypoxia scores captured the influence of the tumor microenvironment on immune response. Meanwhile, KAN-derived features reflected key genomic associated with immunotherapy outcomes. This integration highlights the importance of a multifactorial approach in enhancing the precision of immunotherapy predictions.

Although this study has developed a meaningful approach for predicting immunotherapy response, several issues remain to be solved before its clinical application. While the prediction model demonstrated high accuracy in both the training and test datasets, its practicality still requires further optimization and validation. This study has certain limitations. The training set exhibited an imbalance in the distribution of responders and non-responders, which was addressed using SMOTE. While SMOTE improved the training process by balancing the data, it may have introduced artificial patterns that could affect the model’s performance. Additionally, the small size of the test set could impact the generalizability of the findings.

Integrating this model into clinical practice also presents several challenges, such as the need for patient genomic sequencing, the development of a user-friendly platform for clinicians, and integration with existing clinical electronic workflow systems. The clinical application of this model will require further optimization with more data. Additionally, there are various practical difficulties in the clinical application of this model, such as ensuring its applicability in different hospital settings, optimizing computational resources to reduce costs, and enhancing the performance of predictions. Clinician training is also crucial, as they need to accurately interpret the model’s predictions and combine them with other clinical information for individualized treatment decisions. Therefore, future research should not only focus on optimizing the model. Instead, it should also explore methods to integrate the model into current clinical workflows to improve its practicality and usability.

Furthermore, the biological mechanisms underlying the identified hypoxia-related genes (NOXA) with PD-L1 inhibitors require experimental validation to support their clinical potential. Future research should focus on expanding the dataset, improving data balance without excessive reliance on synthetic methods, and experimentally validating the model. In addition to large-scale cohort validation, in vivo animal and cell model experiments should also be used to further verify the key gene (*NOXA*) in immune regulation. We will explore the detailed regulatory mechanisms of this gene in hypoxia-induced immunotherapy resistance and its potential as a therapeutic target. Meanwhile, prospective clinical trial design is also a crucial direction for further validating the predictive capability of the model. For example, HCC patients receiving PD-L1 inhibitors can be included in the study, where the model is used for prediction, and their actual immunotherapy response is observed. We will further develop a visualization platform to facilitate deployment in real-world settings, translating these research findings into a practical immunotherapy response prediction platform.

## 5. Conclusions

In conclusion, this study provides a novel, integrative approach for predicting PD-L1 inhibitor response in HCC. The KAN model effectively captured the complex relationships between gene expression and immune response. By incorporating hypoxia-related risk features, the model addresses the critical role of the tumor microenvironment in influencing immunotherapy outcomes. This study highlights the importance of multi-dimensional analysis in advancing precision medicine for HCC.

## Figures and Tables

**Figure 1 bioengineering-12-00322-f001:**
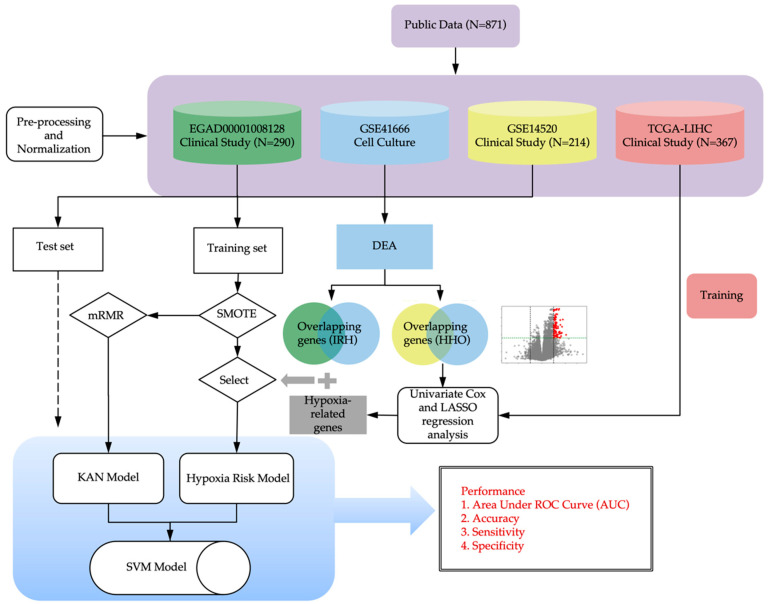
An overview of the computational modeling of immunotherapy response prediction.

**Figure 2 bioengineering-12-00322-f002:**
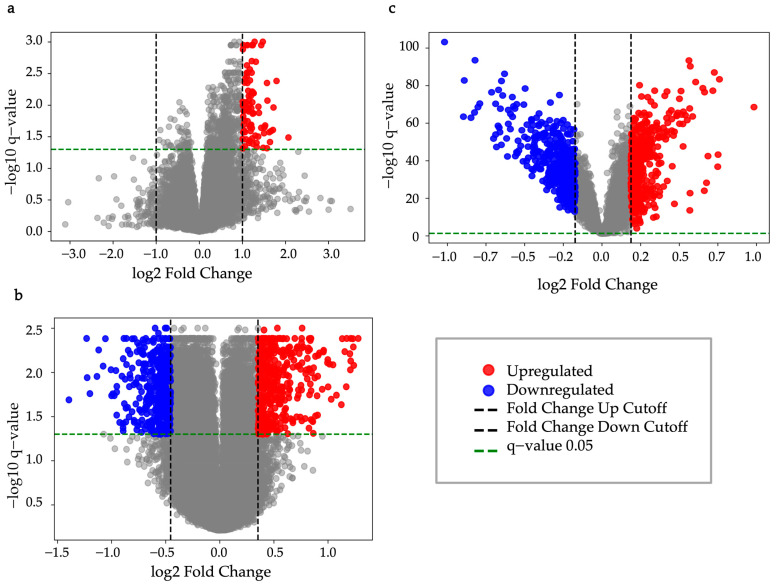
Volcano plot. (**a**) EGAD00001008128, q-value < 0.05, FC > 1.999 (upregulation), and FC < 1/1.999 (downregulation); (**b**) GSE41666, q-value < 0.05, FC > 1.277 (upregulation), and FC < 1/1.370 (downregulation); (**c**) GSE14520, q-value < 0.05, FC > 1.139 (upregulation), and FC < 1/1.1276 (downregulation).

**Figure 3 bioengineering-12-00322-f003:**
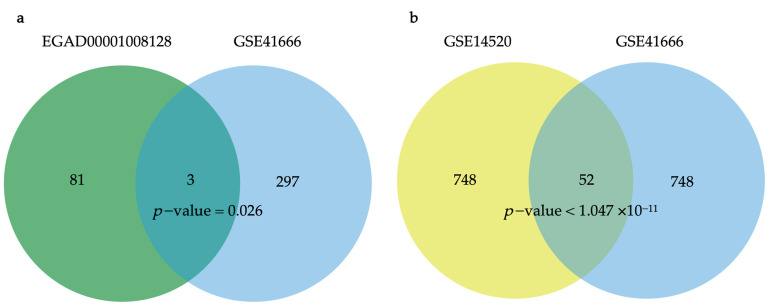
Venn diagrams. (**a**) IRH genes: Overlapping DEGs between EGAD00001008128 and GSE41666 datasets; (**b**) HHO genes: Overlapping DEGs between GSE41666 and GSE14520 datasets.

**Figure 4 bioengineering-12-00322-f004:**
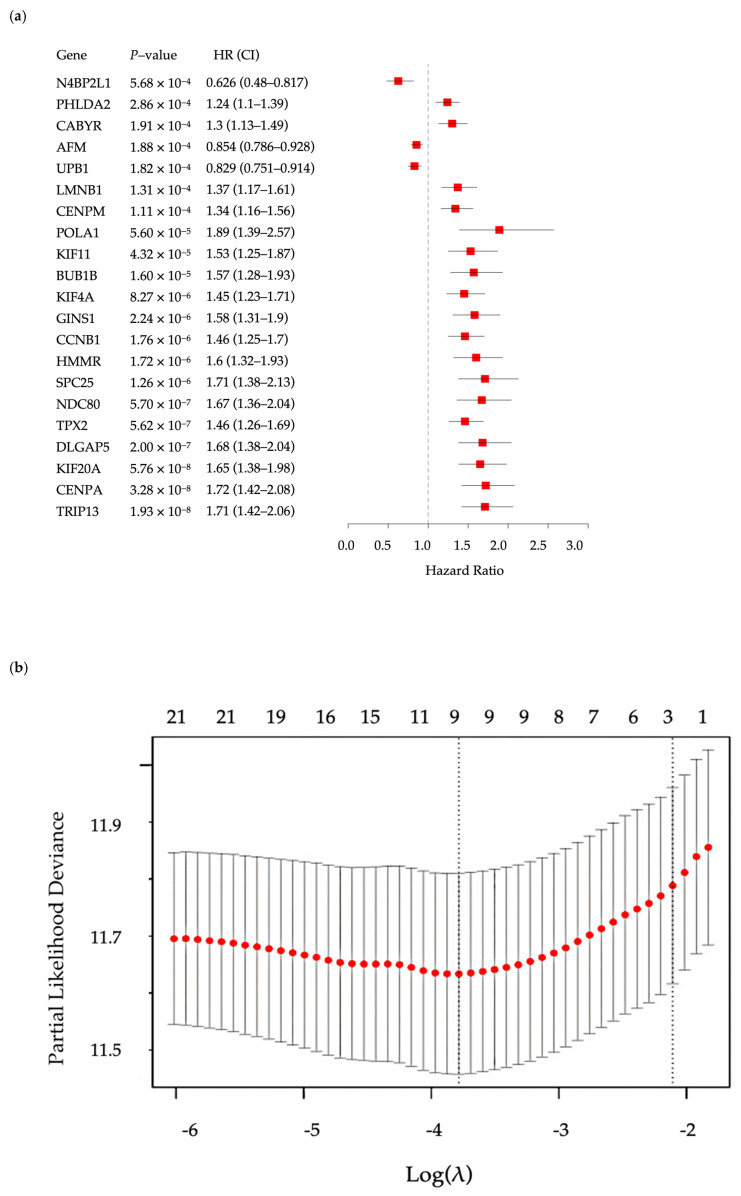
Cox regression and LASSO analysis of 21 candidate genes on HHOs, which resulted in a final model of 9 genes. (**a**) Cox proportional hazard regression; (**b**) Variable selection using Lasso algorithm and the optimal λ selection using cross-validation.

**Figure 5 bioengineering-12-00322-f005:**
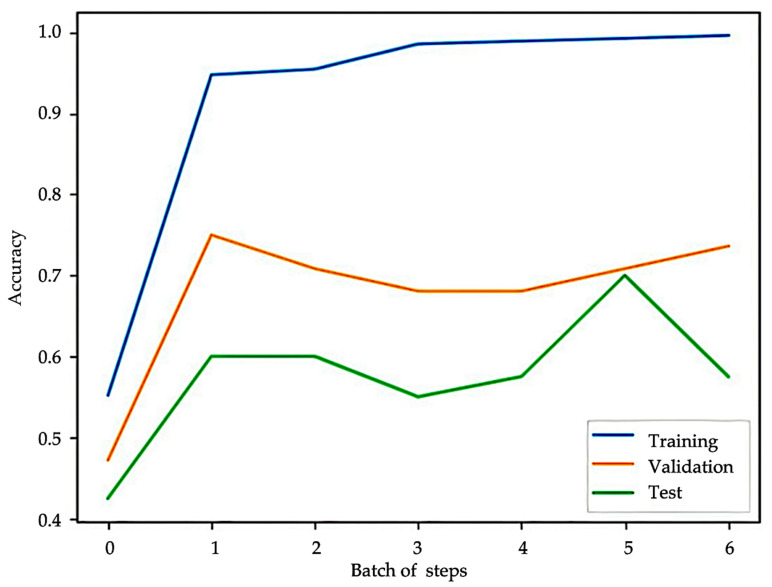
Accuracy comparison of the KAN model: the training accuracy reaches 0.9361, while the validation and test accuracies peak around 0.75 and 0.70, respectively.

**Figure 6 bioengineering-12-00322-f006:**
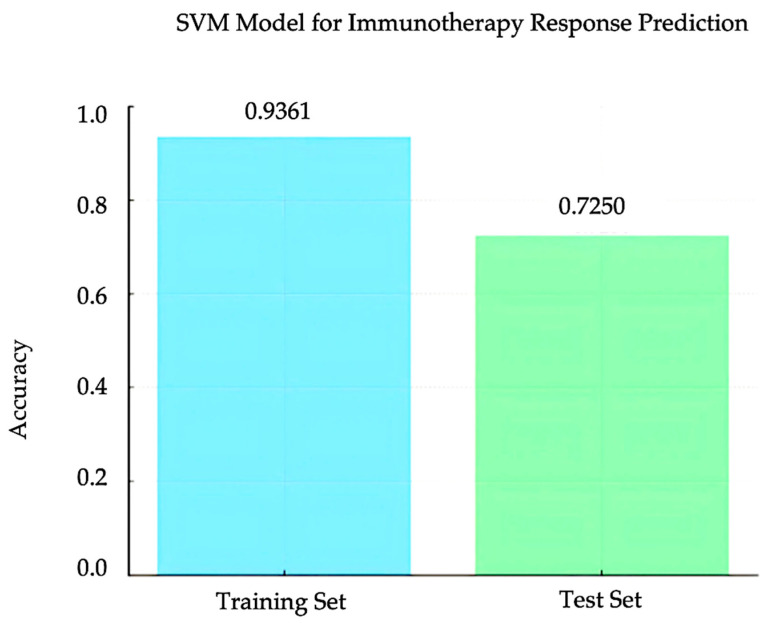
Training and test accuracy for the combined SVM Model. The SVM model achieved 0.9361 accuracy on the training set and 0.7250 on the test set.

**Table 1 bioengineering-12-00322-t001:** Cox regression analysis for the hypoxia risk scoring model related to drug response prediction.

Gene	log HR	log HR SE	HR	t	*p*	95% CI Lower	95%CI Upper
*PHLDA2*	0.1497	0.0753	1.1615	1.9887	0.0467	1.0022	1.3462
*DLGAP5*	0.0757	0.2695	1.0787	0.281	0.7787	0.636	1.8293
*N4BP2L1*	−0.2318	0.1636	0.7931	−1.4163	0.1567	0.5755	1.093
*CENPA*	0.099	0.227	1.104	0.4361	0.6628	0.7076	1.7226
*UPB1*	−0.0584	0.0725	0.9433	−0.8061	0.4202	0.8184	1.0872
*CABYR*	0.1509	0.0752	1.1629	2.0063	0.0448	1.0035	1.3476
*AFM*	−0.0022	0.0609	0.9978	−0.0368	0.9707	0.8856	1.1242
*HMMR*	0.3139	0.1889	1.3687	1.6618	0.0965	0.9452	1.982
*KIF20A*	0.0587	0.2379	1.0604	0.2465	0.8053	0.6652	1.6904
*PMAIP1*	−0.1203	0.1857	0.8866	−0.648	0.517	0.6162	1.2758

**Table 2 bioengineering-12-00322-t002:** Hyperparameters and optimization settings for KAN training.

Parameter	Value
Number of training processes (batches)	6
Number of grid intervals	3–8
Optimizer	L-BFGS algorithm
Learning rate	1
Maximum number of iterations per optimization process	20
Maximal number of function evaluations per optimization process	25
Termination tolerance on first-order optimality	1 × 10^−7^
Termination tolerance on function	1 × 10^−9^

## Data Availability

The data presented in this study are openly available in European Genome-Phenome Archive (https://ega-archive.org/datasets/, accessed on 25 May 2024), National Center for Biotechnology Information database (https://www.ncbi.nlm.nih.gov/, accessed on 25 May 2024), and The Cancer Genome Atlas (https://www.cancerimagingarchive.net/, accessed on 25 May 2024).

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
