# Peer review of "Kolmogorov–Arnold Network Model Integrated with Hypoxia Risk for Predicting PD-L1 Inhibitor Responses in Hepatocellular Carcinoma"

_bioengineering, 2025, doi:10.3390/bioengineering12030322_

Round 1
Reviewer 1 Report
Comments and Suggestions for Authors
The manuscript addresses a timely and essential topic in hepatocellular carcinoma (HCC) research, specifically exploring the use of machine learning models, particularly the Kolmogorov-Arnold Network (KAN), integrated with hypoxia-related factors to predict the response to PD-L1 inhibitors. While the study presents a promising approach, several areas need substantial revision to improve the presentation's clarity, the methodology's depth, and the findings' clinical relevance.
1. The figures in the manuscript are challenging to interpret due to their low resolution and small text. Visual presentation is crucial in a study like this, and as it currently stands, the figures are insufficient for practical data interpretation. Redraw the figures with higher resolution and more extensive, more explicit text to ensure they are legible. In addition, please include explicit markers of statistical significance (e.g., p-values or confidence intervals) within the figures to help readers quickly identify key results.
2. The introduction outlines the background of the study, but the novelty of this research is not sufficiently emphasized. The manuscript should clearly state how this study advances the understanding of predicting immunotherapy response in HCC, particularly in comparison to existing models. You should explain how integrating hypoxia-related factors with the KAN model uniquely contributes to cancer immunotherapy. This will help underline the significance of the study.
3. The methodology section is pretty general and needs more detail. Specifically, how were hypoxia-related genes selected? How was data preprocessing conducted, especially for handling potential data imbalances (such as through SMOTE)? You also mention the use of the KAN model. Still, more explanation is needed about how it compares to traditional methods like MLPs (Multi-Layer Perceptrons), especially in predicting immunotherapy responses. Additionally, more information on how statistical tests were applied and how confounding factors were controlled for would improve the transparency and reproducibility of the study.
4. The clinical implications of the findings are mentioned but not sufficiently explored. You briefly mention how the model might improve early prediction of immunotherapy response in HCC. Still, the discussion would benefit from a more in-depth exploration of how these models could be integrated into clinical practice. Specifically, the manuscript would be more substantial if you addressed the potential challenges to applying this model in real-world healthcare settings, such as the need for clinician training, the integration of machine learning models with existing healthcare systems, and the costs and resource requirements associated with implementing such a model in a clinical workflow.
5. The manuscript cites some relevant references, but many are outdated. The field of immunotherapy and precision medicine in liver cancer has advanced rapidly in recent years. Please update the references, particularly those related to the role of hypoxia in cancer therapy, and incorporate updated studies. This will ensure the manuscript is grounded in the latest research and positioned within the field's current state.
The manuscript is generally understandable, but there are several areas where the clarity and readability of the text could be improved. Some sentences are overly complex, which makes them difficult to follow. Additionally, there are minor grammatical issues and occasional awkward phrasing. Below are specific points that need attention:
1. Some sentences, particularly in the introduction and discussion, are long and complex. Breaking these sentences into shorter ones would make the text easier to follow. For example, "This study uses the Kolmogorov-Arnold Network integrated with hypoxia-related factors to predict the response to PD-L1 inhibitors in hepatocellular carcinoma, which is a highly relevant topic in cancer immunotherapy." could be rephrased as "This study uses the Kolmogorov-Arnold Network (KAN) integrated with hypoxia-related factors. The goal is to predict the response to PD-L1 inhibitors in hepatocellular carcinoma, an important issue in cancer immunotherapy."
2. The manuscript tends to use passive voice in several places, which can make sentences less direct and harder to follow. For example, "The model was trained using the data" can be rephrased as "We trained the model using the data." Using an active voice would make the writing more engaging and more straightforward.
3. There are instances where phrases or ideas are repeated unnecessarily, which can make the writing feel redundant. For example, the phrase "the machine learning model" appears multiple times in close proximity. It can be shortened or substituted with pronouns to avoid repetition.
4. Some terms are overly technical or vague and could be replaced with simpler alternatives for better readability. For example, "significant improvement" could be more specific, such as "statistically significant improvement."
5. Some sentences have minor grammatical issues, particularly with article usage. For example, "This model provides prediction for lung cancer" should be "This model provides a prediction for lung cancer." Similarly, "data preprocessing is a critical step" could be "Data preprocessing is a critical step" for consistency.
Clarity of Key Terms:
The manuscript sometimes uses technical terms without providing clear definitions or context. For instance, abbreviations like "KAN" should be explained fully on first use, especially for readers who may not be familiar with the terminology.
Author Response
Reviewer 1
Comments and Suggestions for Authors
The manuscript addresses a timely and essential topic in hepatocellular carcinoma (HCC) research, specifically exploring the use of machine learning models, particularly the Kolmogorov-Arnold Network (KAN), integrated with hypoxia-related factors to predict the response to PD-L1 inhibitors. While the study presents a promising approach, several areas need substantial revision to improve the presentation's clarity, the methodology's depth, and the findings' clinical relevance.
- The figures in the manuscript are challenging to interpret due to their low resolution and small text. Visual presentation is crucial in a study like this, and as it currently stands, the figures are insufficient for practical data interpretation. Redraw the figures with higher resolution and more extensive, more explicit text to ensure they are legible. In addition, please include explicit markers of statistical significance (e.g., p-values or confidence intervals) within the figures to help readers quickly identify key results.
Response: We have revised all the figures, increasing their resolution to ensure clarity. At the same time, we have enlarged and rearranged the figures to make sure that every piece of text information in the figures is visible. We have marked the p-values either within the corresponding figures or below them, enabling readers to quickly see the key results. The original Figure 2 has been divided into Figure 2 and Figure 3.
- The introduction outlines the background of the study, but the novelty of this research is not sufficiently emphasized. The manuscript should clearly state how this study advances the understanding of predicting immunotherapy response in HCC, particularly in comparison to existing models. You should explain how integrating hypoxia-related factors with the KAN model uniquely contributes to cancer immunotherapy. This will help underline the significance of the study.
Response: We have added the progress of this study compared with existing models in the introduction section. We have elaborated on the uniqueness of integrating hypoxia-related factors into the KAN model, emphasizing that this integration has enhanced the model's predictive ability. The revised content is as follows: “This study used KAN to model the effect of genomic factors on the immunotherapy response in HCC, which facilitates an in-depth exploration of genomic features associated with immune response. Hypoxia plays a critical role in the TME, influencing both tumor immune evasion and treatment resistance. Existing predictive models for HCC immunotherapy response often overlook the interaction between hypoxia and immune regulation. By integrating hypoxia-related factors with KAN, the complex interplay between the TME and high dimensional genomic data has been captured.”
- The methodology section is pretty general and needs more detail. Specifically, how were hypoxia-related genes selected? How was data preprocessing conducted, especially for handling potential data imbalances (such as through SMOTE)? You also mention the use of the KAN model. Still, more explanation is needed about how it compares to traditional methods like MLPs (Multi-Layer Perceptrons), especially in predicting immunotherapy responses. Additionally, more information on how statistical tests were applied and how confounding factors were controlled for would improve the transparency and reproducibility of the study.
Response: Regarding the screening of hypoxia-related genes, we have added detailed processing methods in the revised manuscript. “First, samples with missing survival time or a survival time of 0 days were excluded, resulting in a final set of 367 clinical samples used for training the hypoxia score model. After that, univariate Cox regression analysis was performed on HHOs, followed by using the Least Absolute Shrinkage and Selection Operator (LASSO) algorithm. 10-fold cross-validation was performed using the R glmnet package to determine the optimal parameter λ and the hypoxia-related genes.”
Regarding the issue of data imbalance, we have also supplemented the data processing methods and the principle of the SMOTE. “To address this, our study performed the Synthetic Minority Over-Sampling Technique (SMOTE), one previous study indicated that this approach could improve the performance of the classifier for the minority class[17]. SMOTE was applied as an initial preprocessing step. The SMOTE function from the imbalanced-learn library in Python was used to process the small sample size in the imbalanced dataset. This is achieved by interpolating between an example and its K-nearest neighbors, which could prevent overfitting and enable the classifier to better identify decision boundaries between different classes[18].”
Regarding the comparison between the KAN model and the MLPs, we have also made supplements in the introduction section. “KANs is constructed based on the Kolmogorov-Arnold theorem, using parametrized splines as the learnable activation functions on edges, instead of inputting the weighted sum to the fixed activation functions as Multi-Layer Perceptrons (MLPs). MLPs are feedforward neural networks with multiple layers of neuron connections, which can handle certain nonlinear relationships. But MLPs are always falling into local optima, have limited capabilities in processing high-dimensional data, and have poor interpretability. The modeling performance of KANs is comparable to and, on small-scale tasks, even better than that of MLPs. Compared with traditional deep learning networks, the intuitive representation of KANs facilitates high interpretability, and they can clearly present the mapping relationship between input features and output results[14].”
Regarding the additional parameter results of the KAN model, we have supplemented them in Result 3.3 and added a Table 2 to make the results more intuitive. “In our study, we conducted 6 training rounds, each defined as a batch of steps, with varying numbers of grid intervals ranging from 3 to 8. The number of grid intervals determines the resolution and flexibility of the spline activation functions in the KAN. For model optimization, we employed the L-BFGS algorithm with a learning rate of 1. The maximum number of iterations per optimization process was set to 20, with a maximal function evaluation limit of 25. The termination criteria were defined by a first-order optimality tolerance of 1x10-7 and a function tolerance of 1x10-9 in Table 2.”
- The clinical implications of the findings are mentioned but not sufficiently explored. You briefly mention how the model might improve early prediction of immunotherapy response in HCC. Still, the discussion would benefit from a more in-depth exploration of how these models could be integrated into clinical practice. Specifically, the manuscript would be more substantial if you addressed the potential challenges to applying this model in real-world healthcare settings, such as the need for clinician training, the integration of machine learning models with existing healthcare systems, and the costs and resource requirements associated with implementing such a model in a clinical workflow.
Response: We have supplemented the discussion section with the clinical significance of the model, its future applications, and the challenges it faces. “Although this study has developed a meaningful approach for predicting immuno-therapy response, several issues remain to be solved before its clinical application. While the prediction model demonstrated high accuracy in both the training and test datasets, its practicality still requires further optimization and validation.” “Integrating this model into clinical practice also presents several challenges, such as the need for patient genomic sequencing, the development of a user-friendly platform for clinicians, and integration with existing clinical electronic workflow systems. The clinical application of this model will require further optimization with more data.” “Future research should focus on expanding the dataset, improving data balance without excessive reliance on synthetic methods, and experimentally validating the model. We will further develop a visualization platform to facilitate deployment in real-world settings, translating these research findings into a practical immunotherapy response pre-diction platform.”
- The manuscript cites some relevant references, but many are outdated. The field of immunotherapy and precision medicine in liver cancer has advanced rapidly in recent years. Please update the references, particularly those related to the role of hypoxia in cancer therapy and incorporate updated studies. This will ensure the manuscript is grounded in the latest research and positioned within the field's current state.
Response: We have cited the latest literature in the introduction, discussion, and other sections respectively. The reference numbers are: 4, 6, 7, 9, 12, 14, 15, 17 and 18.
Comments on the Quality of English Language
The manuscript is generally understandable, but there are several areas where the clarity and readability of the text could be improved. Some sentences are overly complex, which makes them difficult to follow. Additionally, there are minor grammatical issues and occasional awkward phrasing. Below are specific points that need attention:
- Some sentences, particularly in the introduction and discussion, are long and complex. Breaking these sentences into shorter ones would make the text easier to follow. For example, "This study uses the Kolmogorov-Arnold Network integrated with hypoxia-related factors to predict the response to PD-L1 inhibitors in hepatocellular carcinoma, which is a highly relevant topic in cancer immunotherapy." could be rephrased as "This study uses the Kolmogorov-Arnold Network (KAN) integrated with hypoxia-related factors. The goal is to predict the response to PD-L1 inhibitors in hepatocellular carcinoma, an important issue in cancer immunotherapy."
Response: We have revised this content to make it easier to read. “We developed a prediction model for PD-L1 inhibitor response in HCC, an important issue in cancer immunotherapy. This study uses the Kolmogorov-Arnold Network (KAN) integrated with hypoxia risk scoring model.”
- The manuscript tends to use passive voice in several places, which can make sentences less direct and harder to follow. For example, "The model was trained using the data" can be rephrased as "We trained the model using the data." Using an active voice would make the writing more engaging and more straightforward.
Response: We have changed the passive voice in many places of the article into the active voice. “In this study, we trained the Cox regression model on multiple publicly available clinical and cell experiment datasets to predict the hypoxia risk.” “Meanwhile, we trained the KAN model on a different clinical dataset to predict immunotherapy response. We used a support vector machine (SVM) to integrate the outputs of two heterogeneous models into a single score.”
- There are instances where phrases or ideas are repeated unnecessarily, which can make the writing feel redundant. For example, the phrase "the machine learning model" appears multiple times in close proximity. It can be shortened or substituted with pronouns to avoid repetition.
Response: We have replaced the repeated words with abbreviations. For example, "machine learning (ML)" and "Kolmogorov Arnold Networks (KANs)". We have ensured that their full names appear only once in the introduction, methods, results, and discussion sections respectively.
- Some terms are overly technical or vague and could be replaced with simpler alternatives for better readability. For example, "significant improvement" could be more specific, such as "statistically significant improvement."
Response: We have changed "significant" to "statistically significant". For example, in Section 3.1, PMAIP1 (NOXA) has been consistently related to hypoxia in previous studies. Further analysis of the GSE41666 and GSE14520 datasets identified 52 statistically significant overlapping genes classified as HCC-Hypoxia Overlap (HHO) genes (p-value = 0.0032) (Figure 3b).
- Some sentences have minor grammatical issues, particularly with article usage. For example, "This model provides prediction for lung cancer" should be "This model provides a prediction for lung cancer." Similarly, "data preprocessing is a critical step" could be "Data preprocessing is a critical step" for consistency.
Response: In the discussion section, we have made changes to these contents. For example, “We developed a prediction model for PD-L1 inhibitor response in HCC by integrating the KAN model with hypoxia risk scoring model.”
Clarity of Key Terms:
The manuscript sometimes uses technical terms without providing clear definitions or context. For instance, abbreviations like "KAN" should be explained fully on first use, especially for readers who may not be familiar with the terminology.
Response: We have added the full name of the Kolmogorov-Arnold Network (KAN) each time it is used for the first time in various parts of the article.
Reviewer 2 Report
Comments and Suggestions for Authors
The authors developed an AI-based model integrating hypoxia-related genomic markers to predict immunotherapy response in hepatocellular carcinoma (HCC) patients. They used publicly available datasets and ML algorithms like Kolmogorov-Arnold Network (KAN) and Support Vector Machine (SVM), the model achieved a test accuracy of 0.725. In general, the main areas for improvement are the presentation of tables and figures. Specific comments are lay out below:
- For the introduction, please consider including a brief comparison of the KAN model's accuracy with other similar models in the field.
- In Figure 2, the figure caption was covered by the bottom of the graph. Please adjust it.
- In Figure 4, the y axis is batch of steps, please state what is the batch size. Please also consider listing all the hyperparameters that are tuned for the model in a table for better illustration.
- Please consider removing Figure 5, the information training set and test set accuracy can just be stated in the text.
Author Response
Reviewer 2
Comments and Suggestions for Authors
The authors developed an AI-based model integrating hypoxia-related genomic markers to predict immunotherapy response in hepatocellular carcinoma (HCC) patients. They used publicly available datasets and ML algorithms like Kolmogorov-Arnold Network (KAN) and Support Vector Machine (SVM), the model achieved a test accuracy of 0.725. In general, the main areas for improvement are the presentation of tables and figures. Specific comments are lay out below:
For the introduction, please consider including a brief comparison of the KAN model's accuracy with other similar models in the field.
Response: We have added a brief comparison between the KAN model and the Multi-Layer Perceptrons (MLPs) in the introduction section. “KANs is constructed based on the Kolmogorov-Arnold theorem, using parametrized splines as the learnable activation functions on edges, instead of inputting the weighted sum to the fixed activation functions as Multi-Layer Perceptrons (MLPs). MLPs are feedforward neural networks with multiple layers of neuron connections, which can handle certain nonlinear relationships. But MLPs are always falling into local optima, have limited capabilities in processing high-dimensional data, and have poor inter-pretability. The modeling performance of KANs is comparable to and, on small-scale tasks, even better than that of MLPs. Compared with traditional deep learning networks, the intuitive representation of KANs facilitates high interpretability, and they can clearly present the mapping relationship between input features and output results[14].”
In Figure 2, the figure caption was covered by the bottom of the graph. Please adjust it.
Response: We have divided Figure 2 into Figure 2 and Figure 3 and placed the label part of the images in the blank area to ensure that the volcano plot is not covered.
In Figure 4, the y axis is batch of steps, please state what is the batch size. Please also consider listing all the hyperparameters that are tuned for the model in a table for better illustration.
Response: Below the original Figure 4 (which has now been changed to Figure 5), we have added Table 2, which shows the training parameters of the KAN model.
Please consider removing Figure 5, the information training set and test set accuracy can just be stated in the text.
Response: Thank you for your feedback. While we acknowledge that the accuracy values can be stated in the text, we believe that Figure 5 (which has been changed to Figure 6) provides a clear visual comparison between the training and test accuracy, making it easier for readers to interpret the generalization ability of the model. Therefore, we respectfully suggest retaining the figure for clarity and accessibility. However, if necessary, we are open to revising the presentation based on the reviewer’s preference.
Round 2
Reviewer 1 Report
Comments and Suggestions for Authors
This revised manuscript presents an intense investigation into using the Kolmogorov-Arnold Network (KAN) combined with hypoxia-related factors to predict PD-L1 inhibitor response in hepatocellular carcinoma (HCC). The improvements made in this revision enhanced the manuscript’s clarity, particularly in the presentation of figures, the methodological details, and the discussion of novelty. However, refinements are still needed to ensure the manuscript is comprehensive and impactful.
The introduction now better explains the study’s novelty, particularly its integration of hypoxia-related gene expression with KAN modeling. However, it could further clarify how this approach improves upon previous immunotherapy prediction models. Adding a comparative discussion with other predictive models would strengthen this section.
The methodology section is more transparent regarding data preprocessing, SMOTE application, and statistical adjustments. However, the rationale for selecting specific hypoxia-related genes remains somewhat unclear. Providing a more detailed explanation of how these genes were prioritized would enhance reproducibility and scientific justification.
The figures have been substantially improved, with better resolution, larger font sizes, and more apparent statistical markers. However, ensuring consistency in formatting and text size across all figures would make data presentation even more effective.
The discussion on clinical implications has been expanded, improving the manuscript’s translational relevance. However, further elaboration on real-world implementation barriers—such as how this predictive model could be integrated into clinical workflows, potential computational challenges, and clinician training requirements—would make this section even stronger.
The future research section now includes specific next steps, such as more considerable cohort validation and additional mechanistic studies, which clarifies how this research will progress. However, proposing experimental validation methods, such as in vivo validation or prospective clinical trial design, would further strengthen this section.
Author Response
Reviewer comments
This revised manuscript presents an intense investigation into using the Kolmogorov-Arnold Network (KAN) combined with hypoxia-related factors to predict PD-L1 inhibitor response in hepatocellular carcinoma (HCC). The improvements made in this revision enhanced the manuscript’s clarity, particularly in the presentation of figures, the methodological details, and the discussion of novelty. However, refinements are still needed to ensure the manuscript is comprehensive and impactful.
- The introduction now better explains the study’s novelty, particularly its integration of hypoxia-related gene expression with KAN modeling. However, it could further clarify how this approach improves upon previous immunotherapy prediction models. Adding a comparative discussion with other predictive models would strengthen this section.
Response: We have added a comparison with other current prediction models in the discussion section. “Traditional immunotherapy predictive models often rely on single biomarkers or clinical features, failing to fully consider the complexity of the TME. For example, tumor mutational burden (TMB)-based predictive models exhibit significant variability in predictive performance across different cancer types and do not account for microenvironmental factors such as hypoxia [27,28].”
- The methodology section is more transparent regarding data preprocessing, SMOTE application, and statistical adjustments. However, the rationale for selecting specific hypoxia-related genes remains somewhat unclear. Providing a more detailed explanation of how these genes were prioritized would enhance reproducibility and scientific justification.
Response: We elaborated the basis for determining hypoxia-related genes in the results section. “Based on the FPKM data from the TCGA-LIHC dataset, these genes were first using univariate Cox regression analysis. Subsequently, LASSO regression was carried out for feature selection to optimize the model and avoid overfitting. Figure 4a shows the results of the Cox proportional hazards regression analysis, including P-values, hazard ratios (HR), and their 95% confidence intervals (CI). The P-values of the 21 candidate genes were relatively low, indicating their statistically significant association with survival. Genes with HR > 1 (e.g., PHLDA2, DLGAP5, CENPA, HMMR, KIF20A) were associated with poorer prognosis, whereas genes with HR < 1 (e.g., N4BP2L1, UPB1, AFM) were associated with better prognosis. Figure 4b shows the LASSO regression variable selection process, where the optimal λ value was determined using cross-validation. By penalizing regression coefficients, LASSO reduced the number of variables and finally selected 9 key genes for the model.”
- The figures have been substantially improved, with better resolution, larger font sizes, and more apparent statistical markers. However, ensuring consistency in formatting and text size across all figures would make data presentation even more effective.
Response: We have revised the text size and format of all the figures and adopted the same format "Palatino" as that of the text part of the journal.
- The discussion on clinical implications has been expanded, improving the manuscript’s translational relevance. However, further elaboration on real-world implementation barriers—such as how this predictive model could be integrated into clinical workflows, potential computational challenges, and clinician training requirements—would make this section even stronger.
Response: We have further supplemented in the discussion section the content regarding the integration into clinical work, existing computational challenges, and clinical doctor training. “Additionally, there are various practical difficulties in the clinical application of this model, such as ensuring its applicability in different hospital settings, optimizing computational resources to reduce costs, and enhancing the performance of predictions. Clinician training is also a crucial, as they need to accurately interpret the model's predictions and combine them with other clinical information for individualized treatment decisions. Therefore, future research should not only focus on optimizing the model. Instead, it should also explore methods to integrate the model into current clinical workflows to improve its practicality and usability.”
- The future research section now includes specific next steps, such as more considerable cohort validation and additional mechanistic studies, which clarifies how this research will progress. However, proposing experimental validation methods, such as in vivo validation or prospective clinical trial design, would further strengthen this section.
Response: We have supplemented the discussion section with content regarding future model validation, experimental validation, prospective research, etc. “In addition to large-scale cohort validation, in vivo animal and cell model experiments should also be used to further verify the key gene (NOXA) in immune regulation. We will explore the detailed regulatory mechanisms of this gene in hypoxia-induced immunotherapy resistance and its potential as a therapeutic target. Meanwhile, prospective clinical trial design is also a crucial direction for further validating the predictive capability of the model. For example, HCC patients receiving PD-L1 inhibitors can be included in the study, where the model is used for prediction, and their actual immunotherapy response is observed.”
